# Aphid Colonisation’s Impact on Photosynthetic and CHN Traits in Three Ornamental Shrubs

**DOI:** 10.3390/insects15090694

**Published:** 2024-09-13

**Authors:** Leopold Poljaković-Pajnik, Nataša Nikolić, Branislav Kovačević, Verica Vasić, Milan Drekić, Saša Orlović, Lazar Kesić

**Affiliations:** 1Institute of Lowland Forestry and Environment, University of Novi Sad, Antona Čehova 13, 21000 Novi Sad, Serbia; branek@uns.ac.rs (B.K.); vericav@uns.ac.rs (V.V.); mdrekic@uns.ac.rs (M.D.); sasao@uns.ac.rs (S.O.); kesic.lazar@uns.ac.rs (L.K.); 2Faculty of Sciences, The Department of Biology and Ecology, University of Novi Sad, Trg Dositeja Obradovića 2, 21000 Novi Sad, Serbia; natasa.nikolic@dbe.uns.ac.rs

**Keywords:** aphids, shrubs, gas exchange, photosynthetic pigments, CHN partition

## Abstract

**Simple Summary:**

This study looked at how three different shrub species in an urban setting were affected by aphid colonisation. Analyses of fourteen factors that describe the photosynthetic pigment concentration and gas exchange to the carbon, nitrogen, and hydrogen partition were conducted. Aphid colonisation had the greatest impact on nitrogen partition and the C/N ratio, parameters that could be recommended in other studies of a similar kind. The studied shrub species had varying responses. *Spiraea × vanhouttei* and *Cydonia japonica* demonstrated the strongest responses in terms of nitrogen partition and the C/N ratio.

**Abstract:**

Shrubs are a significant component of urban vegetation found in parks, but they experience various influences from biotic and abiotic agents, among which aphids play an important role. In this work, the effects of aphid colonisation on three shrub species in urban environments were examined. Fourteen parameters were analysed, describing the photosynthetic pigment content and gas exchange to carbon, nitrogen, and hydrogen partitions. While no significant effect of colonisation was found on photosynthetic pigment parameters, the effect was significant on gas exchange parameters. The strongest effect of aphid colonisation achieved was on nitrogen partition and the C/N ratio, parameters that could be suggested for further similar studies. All parameters were classified into two groups according to their principal component analysis, suggesting a correlation between nitrogen and carbon content, the C/N ratio, measured gas exchange parameters, and chlorophyll a content. The ratio between net photosynthesis and dark respiration (A/K) was classified in the second group, suggesting that this parameter provides additional information on the effect of aphid colonisation and deserves special attention in further studies. There were differences in the effect of aphid colonisation on the physiology of the examined shrub species, especially in cases where a decrease in the C/N ratio was achieved in *Spirea trilobata* and *Cydonia japonica*, while an increase in the same parameter was recorded in *Hybiscus syriacus*.

## 1. Introduction

Vegetation plays an important role in urban environments. Plants inhabit green spaces, such as parks, forests, private residential lawns, and other open areas [1]. These areas are among the primary supporters of sustainability in urban regions, people’s wellness, and urban quality of life; they are considered to provide social, ecological, and economic benefits [1,2]. The increased attention paid to green infrastructure (i.e., the employment of plants in cities) arises from the numerous ecosystem services that they provide, which support and enrich human life [3,4]. The numerous supported or enabled vegetation-based benefits include the production of food, the regulation of noise and air pollution, reductions in the urban heat island effect, the control of biogeochemical cycling, flood mitigation, biodiversity maintenance, promoting the rest and recreation of city residents (beneficial for physical health, socialization, and stress levels), and increasing metropolitan property values [1,5].

Shrubs are a significant component of urban vegetation found in parks, urban wooded settings, gardens, scrubland, hedges, and alongside railway or road infrastructure [5]. The ornamental shrubs *Hibiscus syriacus* L., *Cydonia japonica* (Thunb.) Pers., and *Spiraea × vanhouttei* Vanhoutte are common species of urban ornamental flora in Serbia. Along with other plants in urban ecosystems, they experience various changes due to biotic and natural influences [6]. Speaking of the first influence type, these shrubs were recognized as host plants for various aphid species in previous investigations [7,8,9]. Aphids (Hemiptera, Aphididae) are important worldwide pests, affecting more than 400 cultivated or self-seeding plant species, including ornamental plants [10]. According to a recent report, 374 aphid species are found within the territory of Serbia [9]. Aphids negatively affect their host plants in several ways [11]. Being phloem feeders, they penetrate sieve elements using their stylets to ingest sugars, nitrogen compounds, and other nutrients necessary for plant growth and reproduction. Furthermore, their saliva, when injected during the feeding phase, could have a phytotoxic effect. Along with direct injuries, aphids transmit numerous plant viruses [12]. Aphid colonisation of ornamental plants affects their vigour and aesthetic appearance by promoting the curling, distortion, and chlorosis of leaves; the hardening of the buds; and the malformation and discoloration (fading, stains, necrosis) of various organs [12,13]. Finally, honeydew deposits covering the leaf cuticle are frequently occupied by sooty moulds (black filamentous saprophytic ascomycetes) and may hinder the photosynthetic activity of leaves [11].

Urban shrubs contribute to improvements in environmental quality and human health by decreasing the levels of air pollution and temperature, reducing energy consumption in buildings, intercepting water and reducing surface runoff, and reducing the atmosphere’s carbon dioxide levels (by sequestering carbon, leading to climate change mitigation) [6]. Furthermore, shrubs provide food and shelter for animals [14], contributing to the preservation of and increase in urban biodiversity. Considering the living habits of aphids colonising urban shrubs, as well as the possible negative consequences, it seems necessary to recognize and quantify the changes in the plant attributes underlying plant vigour and fitness in (often inhospitable) urban environments. The adverse effects on plant growth, metabolism, mineral nutrition, photosynthetic rate, concentration of chlorophyll, and gas exchange parameters [15,16,17,18,19,20,21] have been measured and reported in aphid-colonised plants.

The influence of aphids on plant growth elements (diameter and height) has not been quantified in previous research, suggesting that the growth elements are not reliable enough to determine the degree of negative impact of the aphid diet on the host plant. Plant development components are also dependent on other environmental and internal variables that might enable the “concealment” of the aphid diet’s harmful influence [22]. Thus, it appears necessary to investigate and quantify the values of fundamental physiological parameters in colonised and non-colonised plant leaves. The aim of this work was to measure and compare rates of photosynthesis and respiration, as well as the contents of photosynthetic pigments, partitions of nitrogen, carbon, and hydrogen, in healthy and colonised leaves. These results may be helpful for entomologists and plant production specialists, given the significance of identifying and assessing the consequences of aphid assault on host species.

## 2. Materials and Methods

### 2.1. Plant Material

Aiming to compare the physiological characteristics of aphid-colonised and control (non-colonised) leaves, samples of *S. × vanhouttei*, *H. syriacus*, and *C. japonica* were collected in the first half of May, in the early morning. Host plants were growing in the urban environment in green spaces within the campus of the University of Novi Sad (45°14′43″ N, 19°51′05″ E). Along with the examined species, 13 tree species from nine families (*Salicaceae, Ulmaceae, Corylaceae, Tiliaceae, Moraceae, Aceraceae, Rosaceae, Caprifoliaceae,* and *Juglandaceae*) were recorded on the university campus, as well as the following shrub species: *Lonicera nitida*, *Symphoricarpus orbiculatus*, *Lonicera × purpusii*, *Buddleia davidii*, *Physocarpus opulifolius*, *Cotoneaster dammeri*, *Forsyhtia viridissima*, *Mahonia aquifolium*, *Jasminum nudiflorum*, *Physocarpus opulifolius*, *Prunus laurocerasus*, and *Berberis thunbergii*.

Three samples of five fully developed leaves (a total of fifteen leaves per plant) were taken from three colonised and three non-colonised shoot tips. Three plants with almost all their shoots colonised were used to collect the colonised shoot tips, whilst three plants with nearly no shoot colonisation were used to obtain the non-colonised shoot tips. All plants were regularly maintained by pruning, so the branches were 1–2 years old. *H. syriacus* was colonised by *Aphis gossypii*, *C. japonica* by *Aphis spiraecola*, and *S. × vanhouttei* by *Aphis spiraecola*. The dense colonies that covered the colonised shoot tips and leaves are a common feature of both aphid species. The samples were packed into plastic bags and transported in a cool box to the laboratory, where they were stored in a refrigerator for further analysis. Prior to measurements and analyses, insects and their exuviae were gently washed out in a mild water stream and shaken off to remove them from the leaves’ and shoots’ surface. All analyses and measurements were performed in triplicate. Specimens of all the collected aphids were mounted on slides using the Canada balsam technique for determination following [23]. Determination was performed according to Blackman and Eastop [10].

### 2.2. Leaf Photosynthetic Characteristics

The net photosynthesis (A) and dark respiration rates (K) were determined polarographically, using a Hansatech DW1 electrode (Hansatech Instruments Ltd., Norfolk, UK). The quantity of oxygen emitted (mol O_2_ cm^−2^ h^−1^) indicated the level of photosynthesis, while the amount of oxygen absorbed (mol O_2_ cm^−2^ h^−1^) determined the level of respiration. The leaf sections used for gas exchange measurements were cut from the middle of the leaf lamina of both control and colonised leaves, avoiding the leaf nerves. The leaf samples were immersed in a pH 7.6–7.8 buffer solution containing 10 mM NaHCO_3_ [24]. After these measurements, the A/K ratio was calculated. The concentration of chlorophyll a (Chla), chlorophyll b (Chlb), and carotenoids (Car) was measured spectrophotometrically (Beckman DU-65, Brea, CA, USA), in absolute acetone extracts and expressed as mg g^−1^ dry weight [25].

### 2.3. Analysis of Carbon, Nitrogen, and Hydrogen Partition

The elemental content of nitrogen (N), carbon (C), and hydrogen (H) in fully developed, oven-dried leaves of the tested species, both colonised and non-colonised, was measured using the CHN Vario EL III element analyser (Elementar Analysensysteme GmbH, Frankfurt, Germany). The elemental partition of the plant material was determined thermoconductometrically using a standard method [26].

### 2.4. Statistical Analysis

The data were processed using a two-way factorial analysis of variance, with species (three plant host species examined) and colonisation (colonised and non-colonised plants) as the main factors. Tukey’s Honest Significant Difference (HSD) test was then conducted at a significance level of α = 0.05. The relationship between the parameters was described using the Pearson correlation coefficient, based on total means for the species × colonisation interaction. This correlation matrix served as the basis for principal component analysis (PCA). The loadings of the measured parameters on the first two principal components were used to group the parameters, with those having the highest loading on the same component considered correlated and grouped together. Statistical analysis was carried out using the STATISTICA 13.3.1 software package [27].

## 3. Results

### 3.1. Analysis of Variance

According to the results of the analysis of variance in Table 1, there was no significant effect of the factor colonisation or the interaction between species and colonisation on parameters describing the content of photosynthetic pigments. The only significant effect was achieved by the factor species on the variation of Chla, Chla + b, and Car. All parameters describing N, C, and H, as well as the C/N ratio, were significantly dependent on the factor species. For these parameters, a significant effect of colonisation was found for all of them except for H, while the effect of the interaction between species and colonisation was significant only for N and the C/N ratio.

The effects of the factors species and colonisation were significant for all examined physiological parameters, but the interaction between species and colonisation was not significant for any of them.

According to Tukey’s HSD test results, there were no significant differences in the content of photosynthetic pigments between colonised and non-colonised leaves nor in parameters derived from them (Table 2). The lowest content of photosynthetic pigments was found in *C. japonica*. There were no significant differences between the examined species in Chlb and the Chla/Chlb ratio.

In colonised leaves, all parameters describing N or C were significantly lower than in non-colonised leaves, while the difference in hydrogen partition was not significant. In total, the species differed significantly in all examined parameters describing N, C, or H. The highest N was found in *H. syriacus* and the lowest in *S. × vanhouttei*. And the opposite result was obtained in the C/N ratio and H. The highest C was found in *C. japonica* and the lowest in *H. syriacus*. However, at the level of species × colonisation interaction, the reaction of examined species on aphid colonisation was significant only in N and C/N, where higher N and lower values of the C/N ratio were recorded on non-colonised leaves than on colonised leaves. This suggests that these two parameters have a considerable sensitivity to aphid colonisation in the examined shrub species.

In total, all physiological parameters describing dark respiration and photosynthesis differed significantly between colonised and non-colonised leaves. Parameters K and A were significantly lower in colonised leaves, while the A/K ratio was significantly higher, suggesting a higher reaction of leaves on aphid colonisation by K. In total, *H. syriacus* achieved the highest and *C. japonica* the lowest K and A values. However, only in *H. syriacus* was it found that K was significantly lower in colonised leaves than in non-colonised leaves, whilst in other cases, at the level of species × colonisation interaction, the reaction of host species on aphid colonisation by examined physiological parameters was not statistically significant.

### 3.2. Principal Component Analysis

Five principal components were calculated using principal component analysis. Factor loadings between them and the original variables are presented in Table 3. The examined parameters (original variables) had their highest loadings with the first two principal components, which explained 87.1% of the total variance. Because the correlation between principal components is zero, it is assumed that the original variables that have their highest loadings with the same principal component are also correlated between themselves. In this way, the examined parameters are classified into two groups, according to the principal component with which they have their highest factor loading (Figure 1). The first group consists of Chla, N, C, H, C/N ratio, K, and A, and the second group is composed of Chlb, Chla + b, Chla/b ratio, Car, and A/K ratio. According to the results of Tukey’s test, the reaction of host plants to the aphid colonisation was significant for most of the parameters of the first group, except for Chla and H, whereas the reaction for the parameters of the second group was not, except for the A/K ratio. However, along with the poor correlation between traits of the first and second PCA groups, the significance of the second group stresses the fact that the second principal component achieved a relatively high contribution to the total variance (37.6%).

## 4. Discussion

The examined parameters are important indicators of the physiological condition or performance of plants and are the result of the considerably complex process of the reaction of plants on aphid colonisation. Considering the results of this study, aphid colonisation achieved a significant influence on the gas exchange and nitrogen accumulation of host plants.

The results obtained from the examination of photosynthesis and respiration showed some regularities in plant reaction to aphid colonisation. It was noticed that the examined aphid species changes the measured parameters. This is important because parameters by which a plant reacts to aphid colonisation could be used in further studies regarding differences in tolerance to the aphid colonisation between and within woody plant species.

In total, the differences between *C. japonica* and *H. syriacus* in Chla and Chla + b, as well as between *C. japonica* and *S. × vanhouttei* in Chla + b and Car, were significant. However, the reaction of aphid colonisation by parameters describing the content of photosynthetic pigments was not significant, both in total and within each of the host species. Thus, there were no general trends found in the content of photosynthetic pigments as a reaction of host plants on aphid colonisation. This is not in concordance with several studies conducted on wheat [28], Fabaceae species [29,30], sclerophyllous oak species (*Quercus suber* and *Q. ilex*) [30], and soybean [31], which, in general, suggest a decrease in the content of photosynthetic pigments in colonised plants.

According to [32], the ratio of contents of chlorophyll a and chlorophyll b in poplar clones varies between 3:1 and 5:1. Also, it was recorded that in poplar varieties, the ratio is higher in aphid-colonised leaves than in non-colonised leaves [20]. In non-colonised leaves, it varied between 3.21 and 4.11, while in colonised leaves, the ratio was higher and varied between 4.7 and 8.76. Chloroplast degradation is one of the most frequent reactions of plants experiencing aphid feeding, which is indicated by the number of chlorotic spots on leaves [33]. By injection of saliva in plant tissue and especially in vascular bundles, aphids can influence all parts of a plant, including manipulation of plants to induce physiological sinks that are beneficial for their performance [34,35,36].

Lack of statistical significance in traits describing photosynthetic pigment content between colonised and non-colonised plants could be caused by deviations within treatments at the level of species × colonisation interaction, resulting in relatively high residual variation and low precision in detecting significant differences between the treatments. This is probably related to the fact that this study was conducted in field trials, where the start of colonisation was not known. However, it could also indicate the pest tolerance of examined plants, as was the case in studies described by Shahzad et al. [28] and Diaz-Montano et al. [31]. Thus, it seems that traits describing the content of photosynthetic pigments and those derived from them are not suitable for precise analysis of the effects of aphid colonisation in the examined species, especially if the colonisation is mild or plants are tolerant to the aphid colonisation.

However, in total, there was a significant decrease in A and K, as well as an increase in the A/K ratio in colonised plants of the examined species. The decrease in K was particularly clear in *H. syriacus*. The decrease in A and K in colonised leaves is in concordance with the findings of some other studies [20,34], suggesting that changes driven by aphid colonisation are more precise and more likely to be detected by traits describing gas exchange related to photosynthesis than by traits describing the content of photosynthetic pigments.

The influence of aphids on host plants is determined by a variety of factors, including cultivar, plant age, and the developmental stage of the aphid species. A dense aphid colony influences a plant’s physiological state, and it is well known that the photosynthesis process is especially sensitive to such influences. In the colonised plants in our study, a 10–20% decrease in A was recorded, depending on the aphid species and its host, which is in concordance with the results of Miles [33], as well as Shahzad et al. [28], who found a decrease in both the photosynthetic rate and grain yield in several wheat cultivars. At the same time, the decrease in K was even more intense, ranging from 26.5% in *S. × vanhouttei* to 34.4% in colonised *H. syriacus* plants. The fact that the decrease in K is more intense than the decrease in A is confirmed by the increase in the A/K ratio, suggesting that net photosynthesis is favoured against K by plants during aphid colonisation.

By feeding on plants, the aphids take nutrients from phloem sap that are necessary for their growth and development. In our study, aphids caused a significant decrement in nitrogen content and an increase in the C/N ratio in all examined species, in total and within the examined species. In total, the partition of carbon was found to be lower in colonised leaves than in non-colonised leaves, while there were no significant differences in hydrogen partition. However, according to the results of Tukey’s HSD test, differences in the partition of carbon between non-colonised and colonised leaves were not significant within examined species at the level of species × colonisation interaction, whereas they were significant for the N and C/N ratio in all three examined species. These data emphasise the importance of the N partition and the C/N ratio in the evaluation of the reaction of plants to aphid colonisation. Thus, although both N and C belong to the same PCA group as the C/N ratio, the variation in the C/N ratio seems to be more related to N than C. Indeed, it has been confirmed by factorial loadings with principal components, where the rest of the variance in the C/N ratio and N that is not explained by the first principal component is dominantly explained by the second principal component, while that of C is more explained by the third principal component.

Such a close relation between the partition of two of the three examined basic elements, nitrogen and carbon, as well as the C/N ratio with the aphid colonisation, could be of considerable importance for further research in plant reaction on aphid colonisation. However, the second PCA group of parameters also deserves attention in further studies because their low correlation with the first PCA group of parameters and relatively high contribution of the second principal component to the total variance suggest that they carry different and considerable information about the reaction of examined species to aphid colonisation.

We assume that such precise changes in the N partition and C/N ratio are the result of both the plant’s reaction and the effect of aphid colonisation. In phloem sap, the most abundant nitrogen compounds are amino acids, and they are the only nitrogen source for aphids [37]. However, their concentration and composition vary considerably, depending on numerous factors [35]. Leroy et al. [37] present their results on *Acyrthosphon pisum* and *Megoura viciae*-infested *Vicia faba* and cite numerous other studies documenting an increase in the amino acid concentration in phloem and xylem sap because of aphid infestation. They stress that glutamine and asparagine have dominant partitions, while essential amino acids have a relatively small partition. In a study by Jacobs et al. [38], the approximate total amino acid concentration was not attributable to colonisation by *Macrosiphoniella tanacetaria* and *Uroleucon tanaceti* in phloem exudates of young leaves and stems of *Tanacetum vulgare* but only in old leaves of plants colonised by *U. tanaceti*, where a higher total amino acid concentration than in non-colonised plants was found. Züst and Agrawal [35] stress the ability of aphids to manipulate plants to induce physiological sinks, which goes to extremes in symptomatic aphid species that induce senescence and foliar chlorosis, resulting in an increase in the amount of free amino acids in sieve tube sap. Nevertheless, the C/N ratio seems to be more suitable for evaluation of the negative effects of aphid colonisation. The total plant tissue C/N ratio is widely used as an index of the nutritional value of plants for phloem sap-feeding insects [39]. According to Sadras et al. [40], the C/N ratio is the parameter that is more closely associated with the photosynthetic protein content than with the concentration of amino acids in sap. They claimed that the C/N ratio is closely related to plants’ suitability to herbivores, where a high C/N ratio reduces the fitness of insects on host plants. In the same study, negative correlation between the number of *Rhopalosiphum padi* on wheat leaves and the molar concentration of sugars in the stem was found, suggesting that, in addition to lowering N, another important defence mechanism of the plant is the increment in the content of labile carbohydrates, which can lead to osmotic stress in aphids. Both defence mechanisms, a decrease in nitrogen content and increment in phloem sap carbohydrates’ concentration, could contribute to the lower nitrogen content and C/N ratio in colonised than in non-colonised leaves that were found in our study. During evolution, many insects developed mechanisms for improved uptake of nitrogen from plant tissues, since, regarding the insects’ needs, these tissues have a relatively low content of this element [41,42]. Thus, it seems that a decrease in the nitrogen partition and C/N ratio could be directly linked with pests’ ability for the efficient uptake of nitrogen but also with the reaction of the plant to the aphid colonisation. This is in concordance with the significant difference in these parameters between colonised leaves and non-colonised leaves that were found in our study, showing high potential for this parameter to be used in the evaluation of host tolerance to aphid colonisation as well. According to Girousse et al. [43], severe short-term aphid colonisation decreased the shoot elongation of alfalfa plants and carbon deposition in the whole shoot, whereas the nitrogen content was primarily reduced in the apex part of the shoot. Hawkins et al. [44] presented data that suggest an inhibitory effect of aphid colonisation on N and P uptake in some herbaceous leguminous species. In cowpea and pea, the N and P percentage of plant dry weight in colonised plants did not differ from control plants, but because the control plants were greater, the absolute amounts of N and P in the control plants were higher than in colonised ones. However, they recorded the opposite in broad beans, suggesting that the research on the effect of aphid colonisation on nitrogen uptake should be continued. As such, a decrease in N could represent a defence mechanism, which plants use to combat the aphid colonisation as well as the outcome of the aphids’ feeding, reducing the plant’s ability to consume nitrogen.

According to the results of this study, the colonisation of *Aphis gossypii* on *H. syriacus* and *Aphis spiraecola* on *C. japonica* and *Spiraea × vanhouttei* induced no effect on the photosynthetic pigment content, while a significant effect on the gas exchange traits and CHN traits of host plants was recorded. The most precise effect was demonstrated on nitrogen partition and the C/N ratio. A close relationship was found between gas exchange and CHN traits, except for the A/K ratio. Based on the acquired results and previous studies, we assume that the N and C/N ratio, two parameters that are relatively simple to determine, will be important in the rapid assessment of plant species responses to aphids and other sap-sucking insects. Thus, we assume that future research on this topic should give close attention to these criteria.

## Figures and Tables

**Figure 1 insects-15-00694-f001:**
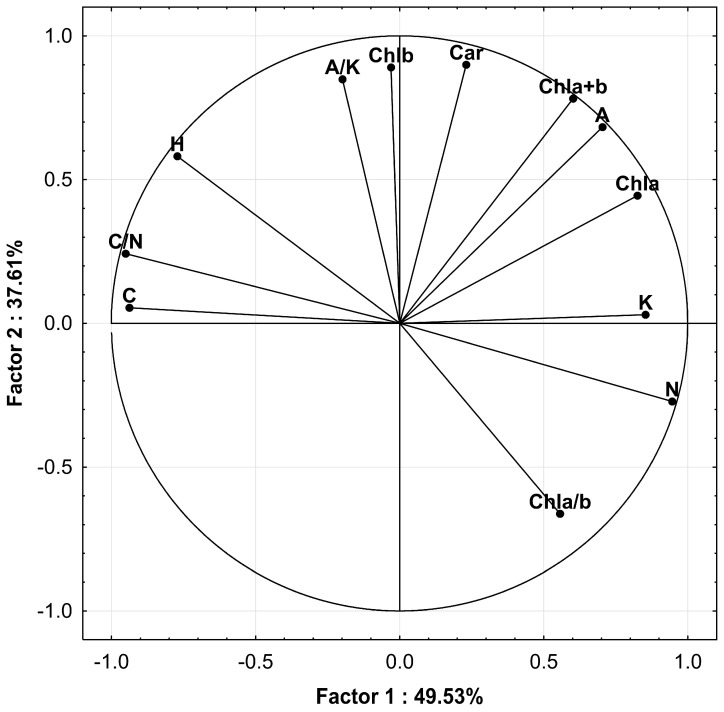
Factor loadings for the first two principal components.

**Table 1 insects-15-00694-t001:** F-test from two-way factorial analysis of variance for examined parameters.

Source of Variation	Chla	^1^	Chlb		Chla + b		Chla/b		Car	
Species (A)	5.286	* ^2^	3.590		7.167	**	2.883		12.341	**
Colonisation (B)	0.978		0.135		0.249		0.141		2.479	
Interaction A × B	2.281		0.680		1.205		0.762		2.920	
	**N**		**C**		**H**		**C/N**			
Species (A)	2330.300	**	295.800	**	72.900	**	17,159.280	**		
Colonisation (B)	251.000	**	5.300	*	0.000		1215.895	**		
Interaction A × B	21.800	**	1.700		2.400		202.505	**		
	**K**		**A**		**A/K**					
Species (A)	22.747	**	22.772	**	13.823	**				
Colonisation (B)	41.592	**	8.354	*	5.951	*				
Interaction A × B	2.257		0.813		0.362					

^1^ Labels of traits: Chla—content of chlorophyll a; Chlb—content of chlorophyll b; Chla + b—content of chlorophyll a + b, Chla/b—Chla/Chlb ratio; Car—content of carotenoids; N—partition of nitrogen; C—partition of carbon; H—partition of hydrogen; C/N—C/N ratio; K—dark respiration; A—net photosynthesis; A/K—A/K ratio. ^2^ Labels for the significance of F-test: *—*p* < 0.05; **—*p* < 0.01.

**Table 2 insects-15-00694-t002:** Tukey’s HSD test for examined parameters in three shrub species with and without aphid colonisation.

Species	Colonisation	Chla	^1^	Chlb		Chla + b		Chla/b		Car	
*S. × vanhouttei*		1.225	ab ^2^	0.694	a	1.919	a	2.130	a	0.672	a
*H. syriacus*		1.430	a	0.460	a	1.890	a	3.440	a	0.490	ab
*C. japonica*		0.926	b	0.313	a	1.239	b	2.996	a	0.322	b
	Non-colonised	1.256	a	0.467	a	1.724	a	2.770	a	0.449	a
	Colonised	1.130	a	0.511	a	1.641	a	2.940	a	0.540	a
*S. × vanhouttei*	Non-colonised	1.418	ab	0.583	a	2.002	a	2.441	a	0.808	a
*S. × vanhouttei*	Colonised	1.031	ab	0.805	a	1.836	a	1.820	a	0.536	ab
*H. syriacus*	Non-colonised	1.549	a	0.515	a	2.064	a	3.153	a	0.524	ab
*H. syriacus*	Colonised	1.310	ab	0.405	a	1.715	a	3.727	a	0.456	b
*C. japonica*	Non-colonised	0.801	b	0.304	a	1.106	a	2.718	a	0.288	b
*C. japonica*	Colonised	1.050	ab	0.321	a	1.371	a	3.274	a	0.355	b
		**N**		**C**		**H**		**C/N**			
*S. × vanhouttei*		2.912	c	47.768	b	6.871	a	16.421	a		
*H. syriacus*		4.331	a	41.801	c	6.331	c	9.658	c		
*C. japonica*		3.049	b	48.764	a	6.656	b	16.072	b		
	Non-colonised	3.579	a	46.401	a	6.622	a	13.465	b		
	Colonised	3.282	b	45.821	b	6.617	a	14.636	a		
*S. × vanhouttei*	Non-colonised	3.006	d	47.855	b	6.905	a	15.921	b		
*S. × vanhouttei*	Colonised	2.818	e	47.682	b	6.836	a	16.922	a		
*H. syriacus*	Non-colonised	4.447	a	41.967	c	6.358	c	9.437	e		
*H. syriacus*	Colonised	4.215	b	41.634	c	6.305	c	9.878	d		
*C. japonica*	Non-colonised	3.284	c	49.381	a	6.602	b	15.037	c		
*C. japonica*	Colonised	2.815	e	48.147	ab	6.710	ab	17.107	a		
		**K**		**A**		**A/K**					
*S. × vanhouttei*		5.822	b	10.184	a	1.755	a				
*H. syriacus*		8.171	a	10.461	a	1.320	b				
*C. japonica*		5.428	b	6.207	b	1.173	b				
	Non-colonised	7.632	a	9.783	a	1.301	b				
	Colonised	5.316	b	8.118	b	1.531	a				
*S. × vanhouttei*	Non-colonised	6.711	b	11.368	a	1.696	ab				
*S. × vanhouttei*	Colonised	4.934	bc	9.000	abc	1.814	a				
*H. syriacus*	Non-colonised	9.868	a	11.447	a	1.169	bc				
*H. syriacus*	Colonised	6.474	bc	9.474	ab	1.472	abc				
*C. japonica*	Non-colonised	6.316	bc	6.533	bc	1.039	c				
*C. japonica*	Colonised	4.539	c	5.882	c	1.307	abc				

^1^ Labels of traits: Chla—content of chlorophyll a [mg g^−1^ FW]; Chlb—content of chlorophyll b [mg g^−1^ FW]; Chla + b—content of chlorophyll a + b [mg g^−1^ FW], Chla/b—Chla/Chlb ratio; Car—content of carotenoids [mg g^−1^ FW]; N—partition of nitrogen [%]; C—partition of carbon [%]; H—partition of hydrogen [%]; C/N—C/N ratio; K—respiration [mol O_2_ cm^−2^ h^−1^]; A—net photosynthesis [mol O_2_ cm^−2^ h^−1^]; A/K—A/K ratio. ^2^ Differences between values with the same letter are not statistically significant at the level α_0.05_.

**Table 3 insects-15-00694-t003:** Factor loadings for five principal components.

Original Variable ^(a)^	Principal Component ^(b)^				
	PC1	PC2	PC3	PC4	PC5
**Chla**	**0.** **825**	0.445	0.009	−0.316	−0.147
**Chlb**	−0.030	**0.** **891**	0.074	0.438	−0.093
**Chla + b**	0.602	**0.** **782**	0.044	−0.015	−0.157
**Chla/b**	0.556	**−0.** **663**	0.325	−0.380	−0.037
**Car**	0.231	**0.** **900**	−0.072	−0.330	0.150
**N**	**0.** **946**	−0.272	−0.014	0.158	0.082
**C**	**−0.** **938**	0.054	−0.315	−0.119	0.059
**H**	**−0.** **772**	0.581	−0.151	−0.207	−0.030
**C/N**	**−0.** **951**	0.242	−0.047	−0.144	−0.116
**K**	**0.** **853**	0.030	−0.518	0.047	−0.031
**A**	**0.** **704**	0.683	−0.146	−0.030	0.128
**A/K**	−0.199	**0.** **849**	0.476	0.036	0.106
**Eigenvalue**	5.944	4.513	0.759	0.653	0.132
**% of the total variance**	49.529	37.609	6.324	5.438	1.100
**Cumulative Eigenvalue**	5.944	10.457	11.215	11.868	12.000
**Cumulative % of the** **total variance**	49.530	87.138	93.462	98.900	100.000

^(a)^ Labels of traits: Chla—content of chlorophyll a; Chlb—content of chlorophyll b; Chla + b—content of chlorophyll a + b, Chla/b—Chla/Chlb ratio; Car—content of carotenoids; N—partition of nitrogen; C—partition of carbon; H—partition of hydrogen; C/N—C/N ratio; K—respiration; A—net photosynthesis; A/K—A/K ratio. ^(b)^ The highest loadings of original variables are bolded.

## Data Availability

The raw data supporting the conclusions of this article will be made available by the authors on request.

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
