# Peer review of "Aphid Colonisation’s Impact on Photosynthetic and CHN Traits in Three Ornamental Shrubs"

_insects, 2024, doi:10.3390/insects15090694_

Round 1

Reviewer 1 Report (Previous Reviewer 1)

Comments and Suggestions for Authors

After the review, the MS “Aphid colonization impact on photosynthetic and CHN traits in three ornamental shrubs” improved. However, there are still some issues that raise doubt. In my opinion, the aim of the work was incorrectly formulated. The 1st point “identify aphid species present in Hibiscus syriacus, Cydonia japonica and Spirea × vanhouttei, ornamental shrub species that are the frequently cultivated in urban areas of Serbia,” is simply material of the research. In the context of the statements included in the modified discussion, the authors tried to assess which of the applied analyses may be useful for determining the tolerance of host plants and/or for rapid assessment of the plant response to aphid feeding. However, in my opinion the main problem is research material. It is too poor (2 species of aphids, 3 species of shrubs, samples taken only once from 1 place) to be able to draw appropriate conclusions.

Other suggestions:

The description of some parts of methodology should be more extensive (lines: 108-109, 112, 118). Full Latin names should be used only for the first mention, the abbreviation should be used in the following text. The same concerns other abbreviations (e.g. N, C, Chla, Car). English proofreading is needed as some sentences are too long and misleading.

More current references concerning aphids feeding on shrubs (trees) should be included in the discussion section.

Detailed comments are also included in the text.

As a whole the manuscript is properly structured and written, but in my opinion the major revision is needed.

Comments on the Quality of English Language

English proofreading is needed as some sentences are too long and misleading.

Author Response

For reviewer 1:

Honorable,

Thank you very much for your suggestions and constructive remarks, which helped us to improve the work. All your comments that are presented in the pdf file we accepted and provided answers in the file “insects-3184550-review 1 Corrections and answers.pdf”. All corrections are provided in the corrected manuscript.

The answers for your general remarks are presented as follows:

After the review, the MS “Aphid colonization impact on photosynthetic and CHN traits in three ornamental shrubs” improved. However, there are still some issues that raise doubt. In my opinion, the aim of the work was incorrectly formulated.

The 1st point “identify aphid species present in Hibiscus syriacus, Cydonia japonica and Spirea × vanhouttei, ornamental shrub species that are the frequently cultivated in urban areas of Serbia,” is simply material of the research.

This aim has been deleted as suggested.

In the context of the statements included in the modified discussion, the authors tried to assess which of the applied analyses may be useful for determining the tolerance of host plants and/or for rapid assessment of the plant response to aphid feeding. However, in my opinion the main problem is research material. It is too poor (2 species of aphids, 3 species of shrubs, samples taken only once from 1 place) to be able to draw appropriate conclusions.

The research is designed in a way to identify traits that are related to the reaction of host on aphid attack, regarding the aim of the research. In that sense, we identified the traits and elaborated our suggestion to propose N partition and C/N ratio to be used in further similar research dealing with evaluation of the reaction of host plants on aphid attack. We assume that this work has the potential to draw attention to these traits as useful ones that could be informative in such studies. Certainly, these results should be further tested in other pests and host species.

Best regards,

Leopold Poljakovic - Pajnik

Reviewer 2 Report (Previous Reviewer 2)

Comments and Suggestions for Authors

Dear authors,

Your manuscript is better now, after revision. This revised form seems better than the original one and I have no other comments.

Kind regards,

R

Author Response

For reviewer 2:

Honorable,

Thank you very much for all your effort in improving our work.

Best regards,

Leopold Poljakovic-Pajnik 

Reviewer 3 Report (Previous Reviewer 3)

Comments and Suggestions for Authors

My previous remarks were sufficiently addressed. 

One thing only:

Line 319 – should be „concentration”

Author Response

For reviewer 3:

Honorable,

Thank you very much for all your effort in improving our work. We corrected Line 319 as you suggested

Best regards,

Leopold Poljakovic-Pajnik 

Round 2

Reviewer 1 Report (Previous Reviewer 1)

Comments and Suggestions for Authors

The article has been corrected according to the comments. 

Comments on the Quality of English Language

Minor English editing recommended.

This manuscript is a resubmission of an earlier submission. The following is a list of the peer review reports and author responses from that submission.

Round 1

Reviewer 1 Report

Comments and Suggestions for Authors

The article presents selected parameters related to photosynthesis and plant respiration as well as the content of photosynthetic pigments, nitrogen, carbon and hydrogen in leaves inhabited and free from aphids taken from three species of ornamental shrubs.

There are several important issues regarding the manuscript. In the current version the manuscript does not meet the requirements for scientific work. It is written sketchy and needs general revision in particular with regards to scientific and English writing The introduction should clearly justify the need for research and indicate the research hypothesis. The first aim of the study was to identify aphid species and characterize their colonies on selected shrub species. However, the results section lacks any data on this issue.

The discussion is written superficially. An in-depth analysis of the results obtained is required. The authors present general and commonly known information on the impact of aphids on host plants.

The correctness of the identification of plants and aphids raises concerns.

The whole manuscript is very casually written The summary must be based on the results obtained and be precise. What is the novelty of the work?

Detailed comments are included in the text.

Comments on the Quality of English Language

English proofreading is mandatory by a native speaker familiar with scientific writing.

Author Response

Dear reviewer,

Thank you for your constructive remarks. We find your suggestion very useful and made appropriate changes to the manuscript. We made major revision of the work to improve quality of the work and English writing. We improved the justification of need for the research as well as indicated the hypothesis. We also provided descriptions of identified aphid species and their colonies.
We improved the section Discussion and provided more detailed and precise analysis of obtained data.
We also made corrections regarding the plants and aphids’ taxonomical identification.
We improved the Summary
Besides corrected manuscript, we are sending explanatory comments within your pdf document.
We hope that you will find our answers and corrections adequate and that our paper will reach the level that is acceptable for publishing.

Kind regards,
Authors

Reviewer 2 Report

Comments and Suggestions for Authors

Dear Author/s,

Suggestions for improvement are as follows:

Line 52: cut point...replace with comma, like that: Hibiscus syriacus, Cydonia japonica, and Spiraea...

Lines 99-111: There is no information related to type of area (park, garden, green space?) and the period in which the observations were made. Or the details of collecting plant samples (shoots, leaves).

The holistic picture of the amount of ornamental plants existing in the researched space (ie on campus) is not clear. It is necessary to mention (even minimally) the existing plant species on campus because the target aphids migrate from one plant to another, being polyphagous, and nearby plants can influence the level of infestation of the 3 species of plants under observation. It should also be clarified if only the 3 plant species were present.

 As such, I suggest adding a small chapter or simply adding requirement informations in actual chapter with those mentioned.

Lines108-109: : Related to: <The level of infestation of all three examined species was high. All the shrubs that are being studied have  shoots colonized on them, which is a trait that Aphis gossypii and Aphis spiraecola have in  common. >...More understandable  would be: All the shrubs studied had shoots colonized, which is a trait common for Aphis gossypii and Aphis spiraecola.

Lines 232-233: Please italicized the scientific names of Cydonia japonica

Lines 300-306: Perhaps you should provide an example from the literature related to one of the aphid species under study (Hibiscus syriacus, Cydonia japonica or Spiraea x vanhouttei) instead of Rhopalosipum which is specific to field plants (maize).

Or if you can't find it, you can make a generalized reference like this (for some species of aphids without mentioning the scientific name). In case of replacement, it is necessary to replace the current Ref or add the reference/s to the Reference List.

Kind regards,

R

Reviewer 3 Report

Comments and Suggestions for Authors

The authors analyzed two photosynthetic parameters and contents of photosynthetic pigments, nitrogen, carbon, and hydrogen in leaves.of three shrubs (Spirea trilobata, Hibiscus syriacus, and Cydonia japonica) which were infested and non-infested by three different species of aphids. The results are interesting and have some value, hovewer manuscript should be substantially improved. In particular, there is no information about aphids (number of specimens, how long they were been feeding, were there any natural enemies of aphids (I am almost sure that there were), dates of taking samples, condition and location of shrubs, part of the canopy from which the leaves were taken, etc. Besides, statistical analysis is not correct – see details below.

Line 52 – should be coma after „…syriaceus”, not dot.

Underline please what is knew in your research

 Lines 104 – 105 – „ Three samples of five fully developed leaves were taken from the tip of shoots of three colonized and three non-colonized mature plants about 10 years old of each examined host species.” – three samples from each plant? Or three in total?

Also compare with the sentence in lines 108 – 109 „All the shrubs that are being studied have shoots colonized on them,…” – these two sentences are in opposite, because in previous one you have written that three samples were taken also from non – colonized plants. Explain this.

Line 123 – what do you mean „the same plants” here?

Line 126 – unnecessary dot before reference

Line 167 – 169 – That there was no effect of colonization on photosynthettic parameters is clear from  Anova  -Table1. In that case there is no need for post-hoc test (Tukey’s HSD) – the same for lines 171 – 172.

According to the Table 1 you got statistically significant differeneces between species – so you should mention that.

Lines 169 – 170 – „The lowest content of photosynthetic pigments was found in Cydonia japonica, and the highest in Hibiscus syriacus and Spirea trilobata.” – but there weren’t any other species, so you can not write that both Hibiscus syriacus and Spirea trilobata has the highest – only one of this species has highest – besides not all differences are statistically significant, so instead of such general sentence describe what was different and what was not.

 Lines 176 – 179 – so – if there wasn’t significant interaction according to Anova in the case of C and H why you have performed Tukey’s test? It shouldn’t be done.

Describe differences which were confirmed statistically – not only mention that they were.

Lines 186 – 190 – Again if Anova shows no significant effect there is no point to do post-hoc test, so this part of Table 2 is also wrong.

 In general, physiological parameters are  in huge part species specific, so better way would be to analyse all these three plants separatelly, especially that you showed statisticall influence of species on almost all parameters.

For sure number of aphids in the colony, for how long they were fedding is also of great importance and could influence all the analysed parameters. More details about the time in season when samples were taken, wheather condition, etc. will be useful as well. Add more details about it.

Moreover, first goal (lines 88 – 90 ) was not entirely achieved – there is nothing about characteristics of colonies of aphids.

Add conclusion, which answered the goals

Comments on the Quality of English Language

English correct
